# ONLINE INTRINSIC REWARDS FOR DECISION MAKING AGENTS FROM LARGE LANGUAGE MODEL FEEDBACK

## ABSTRACT

Automatically synthesizing dense rewards from natural language descriptions is a promising paradigm in reinforcement learning (RL), with applications to sparse reward problems, open-ended exploration, and hierarchical skill design. Recent works have made promising steps by exploiting the prior knowledge of large language models (LLMs). However, these approaches suffer from important limitations: they are either not scalable to problems requiring billions of environment samples, due to requiring LLM annotations for each observation, or they require a diverse offline dataset, which may not exist or be impossible to collect. In this work, we address these limitations through a combination of algorithmic and systems-level contributions. We propose `ONI`, a distributed architecture that simultaneously learns an RL policy and an intrinsic reward function using LLM feedback. Our approach annotates the agent's collected experience via an asynchronous LLM server, which is then distilled into an intrinsic reward model. We explore a range of algorithmic choices for reward modeling with varying complexity, including hashing, classification, and ranking models. By studying their relative tradeoffs, we shed light on questions regarding intrinsic reward design for sparse reward problems. Our approach achieves state-of-the-art performance across a range of challenging, sparse reward tasks from the NetHack Learning Environment in a simple unified process, solely using the agent's gathered experience, without requiring external datasets. We make our code available at URL (coming soon).

## 1 INTRODUCTION

Reward functions are central to reinforcement learning (RL), and are often assumed to be given as part of the problem definition (Sutton & Barto, 2018). These functions are written to describe the task at hand, and often involve tradeoffs between ease of task definition and ease of policy optimization. For example, assigning a reward of $+1$ for solving the task and $0$ otherwise is simple to define and accurately reflects the task goal, but is difficult to optimize due to providing zero gradients almost everywhere.

These difficulties have motivated the use of intrinsic rewards to aid policy optimization (Randlov & Alstrøm, 1998; Ng et al., 1999; Sorg et al., 2010; Singh et al., 2010). The reward designer can include additional reward shaping terms to create a denser learning signal, which can reflect task progress or guide the agent towards intermediate goals. However, designing intrinsic rewards can be remarkably challenging (Booth et al., 2023; Ibrahim et al., 2024) and places increased demands on human experts to provide task-specific knowledge.

Recently, several works have been proposed to leverage the vast prior knowledge encoded in large language models (LLMs) to automate the reward design process, based on a task description in natural language. They can be broadly categorized into two families:

**1. Generating the reward function's code by LLM.** A number of methods have been proposed to automatically generate executable code that computes the reward directly (Ma et al., 2023; Xie et al., 2023; Yu et al., 2023; Li et al., 2024). While they have demonstrated success in complex continuous control tasks, they either require access to environment source code to include in the prompt, or a detailed description of input parameters and reward function templates. Furthermore, they are limited to reward functions compactly expressible via code, describing explicit logic; and

it is unclear how these approaches can easily process high-dimensional state representations such as images, or semantic features such as natural language.

**2. Generating reward values by LLMs.** Motif (Klissarov et al., 2023) is a typical example of this category. It ranks the captions of pairs of observations using an LLM and distills these preferences into a parametric reward model. Motif does not require access to environment source code nor numerical state representation, can process semantic input features, and can scale to problems requiring billions of environment samples. Nevertheless, it also suffers from two important limitations. First, it requires a diverse, pre-existing dataset of captioned observations which are used to elicit preferences from the LLM. In many situations, such a dataset might not exist, and collecting it can increase the sample complexity. More importantly, collecting a diverse dataset often requires a non-trivial reward function that is feasible to optimize, which is the primary problem we aim to solve with intrinsic reward functions in the first place. Second, it involves a complex three-stage process, which sequentially annotates observations using an LLM, trains a reward model, and finally trains an RL agent. This is still time-consuming, given that the LLM annotation process can take several days' worth of GPU hours, and is done prior to training the reward model and RL agent. Alternatively, Chu et al. (2023) query the LLM to directly label observations as having high or low reward at each timestep. However, querying an LLM for every observation is computationally infeasible for many RL applications, which involve millions or billions of observations.

As a consequence, it would be desirable to have *an integrated solution* that offers:

(1) *concurrent and fast online learning of both the intrinsic rewards and the policy that requires no external data nor auxiliary reward functions,*

(2) *expressible reward functions that can capture semantic features that are difficult to process with compact executable code.*

In this work, we present `ONI`, a distributed online intrinsic reward and agent learning system. `ONI` assumes access to captions of observations, similar to previous work (Klissarov et al., 2023; Chu et al., 2023). The captions of collected observations are annotated by an asynchronous LLM server, and both the policy and intrinsic reward model are simultaneously updated using LLM feedback. `ONI` removes the dependency on external datasets beyond the agent's own experience and enables large-scale RL training with ease. Such a learning framework allows us to systematically compare different algorithmic choices for synthesizing LLM feedback. Specifically, we explore three methods: the first one is retrieval-based and simply hashes the annotations; the second builds a binary classification model to distill the sentiment labels returned by the LLM; and the third sends pairs of captions to the LLM server for preference labeling and learns a ranking model, similar to Motif. By carefully comparing the three proposed algorithms, we provide valuable insights into several important questions regarding intrinsic reward design. We demonstrate that `ONI` is able to match Motif's performance across a range of challenging, sparse rewards from the NetHack Learning Environment (NLE) (Küttler et al., 2020), solely using the agent's gathered experience in a single, unified process.

## 2 BACKGROUND

We consider a partially observed Markov decision process (POMDP) setting where the problem is defined by $\mathcal{M} = (\mathcal{S}, \mathcal{A}, \mathcal{O}, p_0, P, O, r, \gamma)$. At each episode, an initial state $s_0 \in \mathcal{S}$ is sampled from the initial state distribution $p_0$. At each time step $t$, the agents observes $o_t \in \mathcal{O}$ which is computed by the emission function $O(s_t)$, and takes an action $a_t \in \mathcal{A}$. This action causes the environment to transition to a new state, $s_{t+1} \sim p(s_t, a_t)$. A new observation $o_{t+1}$ and a reward $r_{t+1} = r(o_{t+1})$ is given to the agent, and the process continues. The goal of the agent is to learn a policy $\pi : \mathcal{O} \to \Delta(\mathcal{A})$ which maximizes the expected return $\mathbb{E}_\pi \left[ \sum_t \gamma^t r_t \right]$. In this work, we additionally assume each observation $o_t$ includes a textual caption $c_t$, which could be empty. For observation spaces without textual captions, $c_t$ could in principle be provided by a captioning model as well.

In many situations, the extrinsic environment reward $r$ is sparse, and the resulting return objective is challenging to optimize. We therefore consider methods which make use of an auxiliary *intrinsic* reward $r^{\text{int}}$ and define a composite surrogate reward:

$$\bar{r}(o_t) = r(o_t) + \beta \cdot r^{\text{int}}(o_t). \tag{1}$$

A key research question is how to define or learn the intrinsic reward $r^{\text{int}}$. A first option is to manually define $r^{\text{int}}$ based on task-specific goals, for example a measure of the distance between

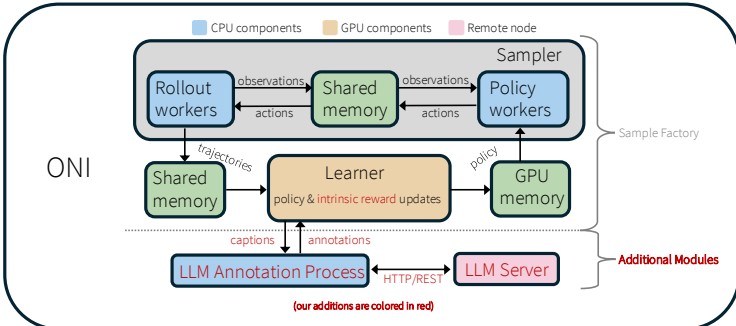

**Figure 3.1:** Overall system diagram of `ONI`. Our additions to Sample Factory are highlighted in red. We added an asynchronously executing LLM server and learned reward function, and connect them back into the main learning process in a way that does not hurt the overall throughput of the policy and value learning.

the agent's current state and the goal state. However, handcrafting the intrinsic reward function can require significant domain knowledge and must be redone for each new task. A second option is to define $r^{\text{int}}$ to measure some notion of observation novelty, which encourages the agent to systematically explore the environment. This can work well in smaller environments, but fails in ones that cannot be exhaustively explored in a tractable amount of time. A third class of methods, which we focus on in this work, leverage LLMs to automatically synthesize $r^{\text{int}}$ to reflect prior knowledge about the task. We discuss all three classes of methods in Section 4.

## 3 ONLINE INTRINSIC REWARDS

### 3.1 SYSTEM DESIGN: DISTRIBUTED PPO WITH LLM ANNOTATIONS

This section outlines the system we have built to learn online intrinsic rewards alongside the policy optimization. The engineering and design here is an important piece of our research, as the rest of our experimental studies are conducted within this system and influenced by the throughput of its interacting components.

Our core system illustrated in Figure 3.1 is built on top of the Sample Factory library v1.0 (Petrenko et al., 2020) and their asynchronous variant of proximal policy optimization (Schulman et al., 2017), referred to as APPO. APPO operates on a single machine and concurrently runs many environment instances while asynchronously updating policy and value estimates with the usual PPO rules, and adequately handles policy staleness and data transfers between these components. Concretely for NetHack, by running 480 environment instances APPO collects approximately 32k environment interactions per second on a Tesla A100-80GB GPU with 48 CPUs.

For online reward learning via LLM annotations in this system[1], we added 1) an LLM server hosted on a separate node, 2) an asynchronously running process that passes observation captions to the LLM server via HTTP request, 3) a hash table that stores the captions and corresponding LLM annotations, and 4) and learning code that dynamically updates a reward model that learns on these annotations. Without being asynchronous, these components have the potential to block the concurrent execution and can reduce the throughput of the overall system, because calling into an LLM and updating the reward model are time-consuming. We added them in a way that retains most of the throughput (!), approximately 80-95% of the original: the average throughput is 30k environment interactions per second if we do not train any additional reward model to distill the LLM annotations, and 26k if a classification-based reward model is learned at the same time (see Section 3.2 for the description of those approaches). The throughput of LLM annotation depends on the actual LLM and prompt we are using. For the reader's reference, when hosting a LLaMA-3.1-8B server on a Tesla A100-80GB GPU using the prompt displayed in Appendix B, `ONI` annotates approximately

---

[1]The prior work by Klissarov et al. (2023) in the offline setting was able to connect their intrinsic reward function into Sample Factory's APPO implementation with a few lines of code, as it just needed to load the PyTorch module of the learned reward function.

810k NetHack captions over the whole training process (2B environment steps, 19-20 hours). The rest of this section overviews the relevant components and how we have modified them.

**Background: APPO's Distributed Execution and Shared Memory**  APPO has three main types of workers:

1. the **learner worker** that coordinates most of the operations, updates the policy and value models, and sends the latest policy ID to the other workers so that they can retrieve it;

2. the **rollout workers** that run copies of the environment, execute actions in them and save the observations; and

3. the **policy workers** that query the policy on new states. The policy workers are separate from the rollout workers so they can efficiently batch across observations.

The workers here are individual processes forked off from a parent process. These all have shared CPU and GPU memory buffers, and efficiently communicate mostly via Python's multiprocessing queues, using pointers to locations in the shared memory. We next describe our new LLM worker, and how we connect it back into the learner.

**Our New Asynchronous LLM Worker and Remote LLM Server**  This worker has input and output queues that communicate back with the learner process, which we use so it does not block the main execution of the system. The LLM worker awaits new observations to label from the input queue, formats them into prompts, calls into an LLM, and returns the annotations in the output queue. Additionally, the prompt templates and other LLM options are configured by the LLM worker. We use an annotation and message format that support all the intrinsic reward labels that we consider in Section 3.2. As Sample Factory already utilizes most of the free CPU and GPU capacity of the system, we opted to call into the LLM via an HTTP/REST interface rather than loading it in this process directly. The communication here adds minimal overhead, and also makes us not need to coordinate the shared memory of multiple GPUs between the main APPO code and LLM. Alongside every APPO run, we allocate an unloaded machine with a new instance of VLLM (Kwon et al., 2023b) to exclusively serve the queries from the RL run. Our system also allows using the same node for both processes if computational resources allow, which further reduces communication costs.

One important design decision is that the asynchronous LLM server is only able to process a small percentage of the overall observations encountered [2], because the environment rollout workers have a much higher throughput than the LLM. An alternative design would be to block the main APPO components and wait for the LLM to label more messages, but we find it more realistic to continue running the policy and label messages as the LLM throughput allows. This also creates the problem of needing to decide on what messages to send to the LLM: the last-in-first-out queue (LIFO), or some uncertainty-based selection. For this work, we use LIFO for simplicity, but note that it would be interesting to investigate alternate approaches in future work.

**Our Modified Learner**  Lastly, we connect this new LLM worker back into the main learner, and dynamically learn a reward model on the annotations obtained from it. To do this, we modified the two threads of the learner worker:

1. the **training thread** is the main thread that a) aggregates the new trajectories from the latest environment interactions; b) decides which new observations to send to the LLM; c) updates the policy, value, and (now) our intrinsic reward model.

2. The **feedback processing thread** (in fact, the initial thread of the learner worker) receives the latest annotations from the LLM worker and updates the dataset (or hash table) that the reward model is trained on. This thread also initializes the reward model at the beginning.

To prevent overfitting the limited amount of annotations in the early stage of training, the training thread only continuously updates the reward model after we receive 25k annotations. Before that, the feedback processing thread run a few updates of the reward model every time we receive new annotations. Now that we have a flexible way of annotating messages and fitting a reward model on top of them, we turn to defining the types of intrinsic reward functions.

---

[2]often $\leq 0.04\%$ for a LLaMA-3.1-8B instance using a single A100-80GB GPU, or $\leq 0.02\%$ when using a V100-32GB GPU

## 3.2 Intrinsic Reward Functions

`ONI` offers flexible algorithmic choices for querying an LLM and distilling its feedback. In this work, we consider the following three methods.

**Retrieval** The simplest approach we consider uses binary labeling and retrieval. The LLM is asked to assign a binary label $y_i \in \{0, 1\}$ indicating whether a caption $c_i$ is "helpful" or "unhelpful" for making progress on the task. The learner worker maintains a hash table $\mathcal{H}$ to store labeled pairs $(c_i, y_i)$, managed by the feedback processing thread. In the training thread, each time the RL agent receives an observation $o_t$ with caption $c_t$, we check if $c_t \in \mathcal{H}$ and the intrinsic reward is defined as:

$$r^{\text{int}}(o_t) = \begin{cases} \mathcal{H}(c_t) & \text{if } c_t \in \mathcal{H} \\ 0 & \text{if } c_t \notin \mathcal{H} \end{cases} \tag{2}$$

If $c_t$ is unlabeled, it is placed into a last-in-first-out (LIFO) queue $Q$ managed by the LLM annotation process, and then sent to the LLM server. The LLM continuously processes elements $c_i$ from $Q$ and returns their labels $y_i$ to the data processing thread of the learner worker, where the pairs $(c_i, y_i)$ are added into $\mathcal{H}$. As described in Section 3.1, the training thread and the feedback processing thread run asynchronously. Therefore, editing the hash table does not slow down policy training. This retrieval-based approach does not generalize to observations with unlabeled captions. However, the resulting intrinsic reward is simple and hyperparameter-free, and may work well when the set of captions belongs to a relatively small set.

**Classification** The second approach we consider is based on binary labeling together with training a classification model. Similarly to above, we label observation captions $c_i$ with labels $y_i \in \{0, 1\}$ indicating whether they are helpful or unhelpful via LLM. We simultaneously train a binary classification model to predict $y_i$ from $o_i$. More precisely, we model $P(y = 1|o)$ by $r_\phi^{\text{int}} : \mathcal{O} \to [0, 1]$, which is then used to compute the binary intrinsic reward by thresholding it at $\eta$:

$$r^{\text{int}}(o_t) = \mathbb{I}[r_\phi^{\text{int}}(o_t) > \eta], \tag{3}$$

where $\mathbb{I}$ is the indicator function. We study the impact of $\eta$ in Section 5.3. Unlike the previous approach, this method has potential to generalize to observations whose captions are similar, but not identical, to the captions labeled by the LLM. However, like the previous approach, it will assign a same reward to observations which are slightly positive (such as finding a few gold pieces in NetHack) and very positive (finding hundreds of gold pieces or a rare artifact).

**Ranking** The third approach we consider is based on ranking observations via pairwise classification, which is the approach taken by Motif. Here, pairs of observations $(o_1, o_2)$ are sent to the LLM, which returns a preference label $y \in \{1, 2, \varnothing\}$ indicating whether $o_1$ or $o_2$ is more desirable for accomplishing the task, or if they are equivalent. A reward model $r_\phi^{\text{int}} : \mathcal{O} \to \mathbb{R}$ is trained by minimizing the negative log-likelihood :

$$-\mathbb{E}_{(o_1, o_2, y) \sim \mathcal{H}} \left[ \left( \mathbb{I}[y = 1] + \tfrac{1}{2}\mathbb{I}[y = \varnothing] \right) \log P_\phi(o_1 \succ o_2) + \left( \mathbb{I}[y = 2] + \tfrac{1}{2}\mathbb{I}[y = \varnothing] \right) \log P_\phi(o_1 \prec o_2) \right] \tag{4}$$

where we use average log-likelihood when $y = \varnothing$[3] and use the Bradley-Terry model (Bradley & Terry, 1952) $P_\phi(o_1 \succ o_2) = 1 - P_\phi(o_1 \prec o_2) = \exp\left(r_\phi^{\text{int}}(o_1)\right) / \left[ \exp\left(r_\phi^{\text{int}}(o_1)\right) + \exp\left(r_\phi^{\text{int}}(o_2)\right) \right]$. Motif computes the mean $\mu_{\mathcal{D}}$, standard deviation $\sigma_{\mathcal{D}}$, and a fixed quantile $\nu_{\mathcal{D}}$ of $r_\phi^{\text{int}}$ over the offline dataset of annotations $\mathcal{D}$. During RL training, it normalizes and thresholds $r_\phi^{\text{int}}$ to give the reward

$$r_{\text{motif}}^{\text{int}}(o_t) = \mathbb{I}[(r_\phi^{\text{int}}(o_t) - \mu_{\mathcal{D}})/\sigma_{\mathcal{D}} > \nu_{\mathcal{D}}] \cdot (r_\phi^{\text{int}}(o_t) - \mu_{\mathcal{D}})/\sigma_{\mathcal{D}} \tag{5}$$

For `ONI`-ranking, applying these steps directly is not possible since the annotation dataset is continuously changing. We thus replace $(\mu_{\mathcal{D}}, \sigma_{\mathcal{D}})$ by a running mean and standard deviation $(\mu, \sigma)$ computed over the experience, and replace the quantile $\nu_{\mathcal{D}}$ by a quantile of the standard normal.

## 4 Related Work

**Exploration Based on Novelty Bonuses** Learning from sparse or otherwise difficult-to-optimize reward functions is a long-standing problem in reinforcement learning. There is a large body of

---

[3]In the equivalent cross entropy minimization formulation, this amounts to using a uniform target $[\tfrac{1}{2}, \tfrac{1}{2}]$ when $y = \varnothing$.

work which defines intrinsic rewards based on novelty bonuses (Schmidhuber, 1991; Kearns & Singh, 2002; Brafman & Tennenholtz, 2002; Stadie et al., 2015; Bellemare et al., 2016; Pathak et al., 2017; Burda et al., 2019; Shyam et al., 2019; Raileanu & Rocktäschel, 2020; Ecoffet et al., 2019; Agarwal et al., 2020; Zhang et al., 2021; Henaff et al., 2022a; Lu et al., 2024). These methods tend to make minimal assumptions about the task at hand, operate online without requiring external data, and sometimes come with theoretical guarantees. However, since they are fundamentally *tabula-rasa*, they must rediscover much of the structure in the task that might already be encoded as prior knowledge in an LLM. Therefore, they tend to have difficulty exploring environments of very high complexity, such as NetHack, in a tractable amount of time (Klissarov et al., 2023).

**LLM-aided Reward Design** In addition to Motif (Klissarov et al., 2023), several works have sought to leverage the prior knowledge encoded in LLMs to produce intrinsic rewards. Eureka (Ma et al., 2023), Auto-MC (Li et al., 2024), L2R (Yu et al., 2023) and Text2Reward (Xie et al., 2023) all use LLMs to generate executable code which computes intrinsic rewards from the under-lying environment state, conditioned on a task description. The generated reward function code is then iteratively improved based on aggregate statistics from agents trained with the current reward. However, a disadvantage with intrinsic rewards represented as code is that they require access to an interpretable underlying state representation, and it is unclear how to leverage non-numerical fea-tures such as those provided by unstructured language captions. The works of Kwon et al. (2023a); Chu et al. (2023) also successfully used LLMs conditioned on task descriptions to directly generate binary rewards in an online manner, and did not train a reward model. This was possible due to eval-uating on environments and tasks which could be solved with a relatively small number of observa-tions, whereas we consider complex open-ended environments with billions of observations so that labeling them all with an LLM would be computationally infeasible. Also of note is the work of Wu et al. (2024), which additionally conditioned LLMs on user manuals to define the intrinsic reward.

**Goal-conditioned Reward Design** A different approach to reward function design is to define re-wards as the distance between the agent's current state and the goal. For example, one line of work learns a state embedding in a self-supervised manner which converts geodesic distances in the original space to Euclidean distances in feature space (Wu et al., 2019; Wang et al., 2021; Gomez et al., 2024). Another line of work of (Fan et al., 2022; Rocamonde et al., 2023; Adeniji et al., 2023; Kim et al., 2024) leverages pretrained image and text encoders, and defines rewards to be some measure of similarity between embeddings of visual observations and embeddings of textual task descriptions. Using a question generation and answering system, Carta et al. (2022) extract auxiliary objectives from the goal description and construct intrinsic rewards. An interesting com-bination of goal-conditioned and LLM-aided reward design is the `ELLM` approach introduced in Du et al. (2023b), which generates candidate goals and uses the distance in LLM embedding space to define the reward. This approach shares the limitations of the works of Kwon et al. (2023a); Chu et al. (2023) discussed in the previous section, in that it requires an LLM call for each agent obser-vation, which becomes computationally infeasible in high-throughput settings involving billions of observations. We discuss more works that utilize LLM for RL broadly in Appendix D.

## 5 EXPERIMENTS

**Environment** We use the NetHack Learning Environment (NLE) (Küttler et al., 2020) as our experimental testbed, since it is one of the most challenging open-ended, long horizon and sparse reward environments available, and was also used as the main environment in the prior work we compare to. NetHack is a classic dungeon crawling game which presents a number of interesting challenges for RL agents: it is procedurally generated, requiring generalization; rewards are sparse for most tasks, requiring exploration; the environment is partially rather than fully observable; transitions are naturally stochastic; episodes are very long, requiring tens to hundreds of thousands of steps to win the game; the dynamics are highly complex, involving large numbers monsters, objects, non-player characters and other entities. Succeeding in the game requires mastering and balancing diverse behaviors including exploration, resource management, object use, combat, puzzle solving and skill progression.

**Tasks and Metrics** We evaluate our agents in two ways: how well they are able to succeed in tasks which have a (typically sparse) extrinsic reward, and how well they are able to progress in the game using the intrinsic reward only. For the former, we use one dense reward task and three sparse reward tasks used in prior work (Küttler et al., 2020; Klissarov et al., 2023), listed below.

1. The `Score` task treats the in-game score[4] as a dense extrinsic reward.

2. The `Oracle` task requires finding the in-game Oracle character, which resides deep in the dungeon. The agent receives a reward of 50 if it manages to reach the Oracle, zero otherwise.

3. The `StaircaseLvl3` and `StaircaseLvl4` tasks requires reaching the third or fourth level staircase and zero otherwise—this requires exploring multiple levels in order to find staircases which lead deeper into the dungeon, while fighting or escaping monsters to survive.

Previous work found that the `Oracle` task can be solved in unexpected ways via reward hacking; however, this is still a challenging sparse reward problem and we include it for completeness. We also train agents with the intrinsic reward only and measure the game progress via four metrics:

*experience level, dungeon level, gold,* and *scout (number of unique locations explored).*

Removing the extrinsic reward gives us a clearer picture of what the intrinsic rewards are prioritizing.

**Methods and Hyperparameters** We instantiate `ONI` with the three reward functions described in Section 3.2. We name the three approaches `ONI-retrieval`, `ONI-classification` and `ONI-ranking`, respectively. For policy learning, we use the Chaotic Dwarven GPT5 (CDGPT5) architecture (Myffili, 2021) used in prior work (Piterbarg et al., 2023; Kurenkov et al., 2023; Klissarov et al., 2023). Architecture details can be found in Appendix A.1. All the methods are trained with two billion ($2 \times 10^9$) environment steps. We train `ONI-classification` with with classification threshold $\eta = 0.5$, where the intrinsic reward coefficient is $\beta = 0.1$ for the `Score` task and $\beta = 0.4$ for all the other sparse-reward tasks and the reward-free agent. Similarly, `ONI-retrieval` uses $\beta = 0.1$ for `Score` and $\beta = 0.5$ for the others. `ONI-ranking` is trained with $\beta = 0.05$. Full details of the training process can be found in Appendix A.2.

**LLMs** We use the LLaMA-3 herd of models (Dubey et al., 2024) as our LLMs. All prompts are listed in Appendix B. We initially reran the Motif baseline using the official code[5] and compared the performance of LLaMA-3.1-70B-Instruct and LLaMA-3.1-8B-Instruct (see Appendix C.1). We did not observe a significant difference in their performance on the `Oracle` or `Score` tasks. Therefore, we use LLaMA-3.1-8B-Instruct in our subsequent experiments to reduce computation.

**Baselines** We compare to agents trained with extrinsic reward alone, Motif (Klissarov et al., 2023) , and a variant of the `ELLM` algorithm (Du et al., 2023a). It is not feasible to directly apply `ELLM` to our setting, since it requires an LLM call for each observation, where the total number of calls scales to the billions in our case. Therefore, we designed a more scalable variant which i) replaces the LLM embedding with a lightweight bag-of-words embedding [6] using FastText (Bojanowski et al., 2016), and ii) includes an episodic term in the intrinsic reward, as with Motif and `ONI` (see Appendix A.2). We call it `ELLM-BoW` and include more details in Appendix A.3. We note that Klissarov et al. (2023) found that other exploration methods based on novelty bonuses such as RND (Burda et al., 2019), NovelD (Zhang et al., 2021) and E3B (Henaff et al., 2022a) did not improve over the extrinsic reward baseline on these tasks, hence we do not include them.

## 5.1 MAIN RESULTS

**Task Performance** We report the average performance and $95\%$ confidence intervals computed via standard errors over 5 seeds in Figure 5.1a. The extrinsic reward agent performs reasonably well on the dense `Score` task, but completely fails on the others due to reward sparsity. We find that, despite not requiring any external data, our `ONI-classification` agent is able to match Motif on all tasks except `Oracle`, where it still performs well. We note that Motif requires pre-collecting data from the `Score` task even for sparse reward tasks—this assumes access to an additional dense reward function, which is an often unrealistic assumption. It also incurs additional one billion ($10^9$) environment samples prior to policy training. All of our agents significantly outperform the extrinsic only baseline on the sparse reward tasks, demonstrating they are able to explore effectively while synthesizing their intrinsic rewards from online data alone.

---

[4]https://nethackwiki.com/wiki/Score

[5]https://github.com/facebookresearch/motif

[6]We also tried using Sentence Transformers (SBERT) for generating sentence embeddings, which led to an approximately 20x throughput drop: 28k vs 1.4k FPS on a Tesla V100-32GB GPU, when the sentence embeddings can be cached. If we recompute the sentence embedding in the forward pass every time, the throughput drop is approximately 100x.

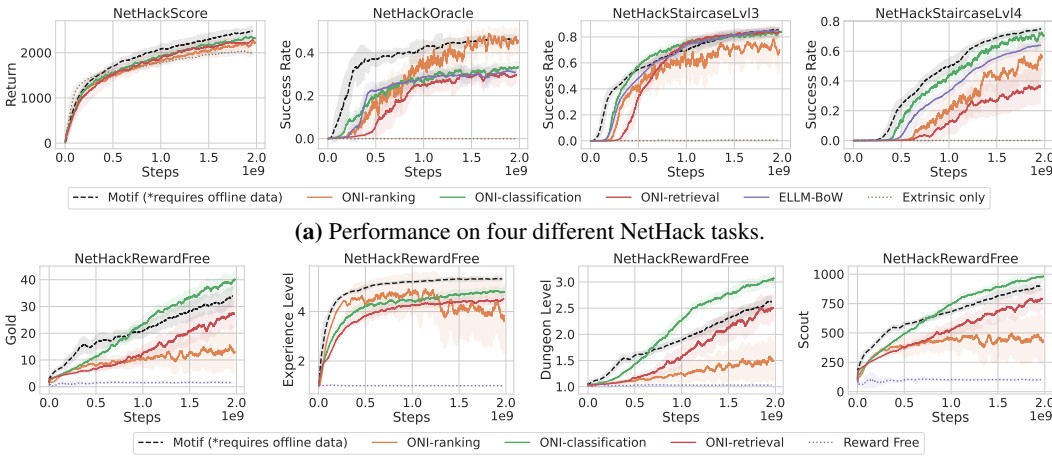

**(a)** Performance on four different NetHack tasks.

**(b)** Game progress of intrinsic rewards only agents.

**Figure 5.1:** `ONI`-based methods are able to match or closely track the performance of Motif without using an pre-collected dataset. This includes (a) reward-based and (b) reward-free settings. Motif's pre-collected dataset uses privileged information about dense reward functions to solve sparse-reward or reward-free environments while `ONI`-methods do not. `ELLM-BoW` demonstrated to be a competitive baseline here too.

Despite its simplicity, `ONI-retrieval` performs surprisingly well, and its performance is often close to that of `ONI-classification`. This is likely a consequence of many messages with positive valence being repeated in the early game of NetHack, such as "*You find a hidden passage*" that allows exploring the rest of the level, or "*You kill the* {monster}!" which leads to experience gain. `ONI-classification`, which also predicts binary rewards but is able to generalize to unseen messages, provides a modest but consistent improvement over `ONI-retrieval` across all environments. This suggests that learning a reward model is indeed helpful. We would expect this gap to increase in settings with added noise or caption diversity, since they increase the likelihood of observed captions being unique.

We do not observe any significant gains from using `ONI-ranking` over `ONI-classification`, despite it being more conceptually general and able to represent a continuous range of intrinsic rewards rather than binary values. This may be because our tasks take place at the very earliest part of the game of NetHack, where only a small fraction of all possible messages are observed, which would also explain the relatively strong performance of `ONI-retrieval`. We hypothesize that more benefits will appear in settings with higher observation diversity.

`ELLM-BoW` performs surprisingly well on these tasks, closely tracking or matching Motif and `ONI` methods. However, in Section 5.2 we highlight a fundamental limitation of `ELLM-BoW`, namely its inability to capture complex semantic meaning, whereas `ONI` is capable thanks to its use of an LLM. It is worth noting that directly using `ELLM-BoW` as in Du et al. (2023a) without our episodic bonus completely fails, see Appendix C.7.

**Game Progress of Intrinsic-Reward-Only Agents** Results for all methods trained in the reward-free setting are shown in Figure 5.1b. All of our `ONI` methods are able to make meaningful progress across all metrics. Interestingly, different variants appear to prioritize different forms of progress: `ONI-ranking` performs best in terms of experience level, whereas `ONI-classification` performs best according to the other metrics. We include the results of Motif for reference, yet emphasize it is actually inapplicable in a truly reward-free setup since it assumes access to dense rewards.

## 5.2 COMPARISONS FOR MORE COMPLEX GOALS

Even though `ELLM-BoW` performs well in Section 5.1, we have found that it does not capture the semantic meanings of more complex goal strings due to the simple bag-of-word representation. To demonstrate this, we train `ELLM-BoW` and `ONI-retrieval` for two opposite goals in the extrinsic-reward-free environment: **(Gold)** "collect gold but do not kill monsters" vs **(Combat)** "kill monsters but do not collect gold". Figure 5.2 shows that `ELLM-BoW` produces two agents of nearly identical

behavior in terms of collected gold and killed monsters. In contrast, `ONI-retrieval` is able to distinguish the two goals and produces agents that emphasize either combat engagement or gold collection, depending on the goal string. See Appendix C.6 for the goal strings for `ELLM-BoW` and prompts for `ONI-retrieval`, respectively.

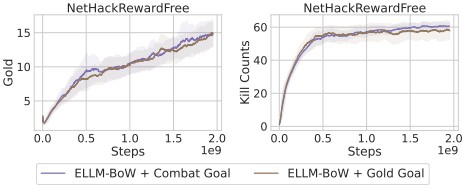 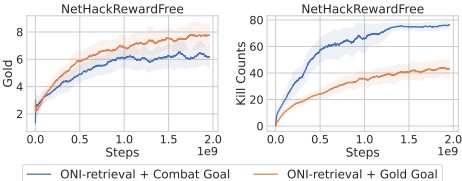

**(a)** `ELLM-BoW` under 2 different goals     **(b)** `ONI-classification` under 2 different goals

**Figure 5.2:** **(a)** `ELLM-BoW` is not able to understand the semantic meaning of complex goals, resulting in agents with similar behavior under the combat and the gold goal. **(b)** `ONI-retrieval` can distinguish the goals and the resulting agents focus on different aspects of game progress.

### 5.3 ABLATION STUDY

**`ONI-classification`: the impact of the classification threshold** As described in Section 3.2, `ONI-classification` predicts binary rewards by modeling $P(y_t = 1|o_t)$ and then thresholding with $\eta$. Instead of using binary reward, an alternative design choice is to use real valued reward

$$r^{\text{int}}(o_t) = P(y_t = 1|o_t), \tag{6}$$

where $r^{\text{int}}(o_t) \in [0, 1]$. is the output of the reward classifier before thresholding. Figure 5.3 shows that the performance of `ONI-classification` on the four NetHack tasks is relatively robust to this hyperparameter $\eta$, and using $P(y_t = 1|o_t)$ as the reward leads to similar performance. As the training progresses, our reward model outputs values close to 0 or 1[7], and we hypothesize this is the reason why the final performance remains comparable. Moreover, the natural classification threshold $\eta = 0.5$ marginally outperforms the other values in the reward-free setting.

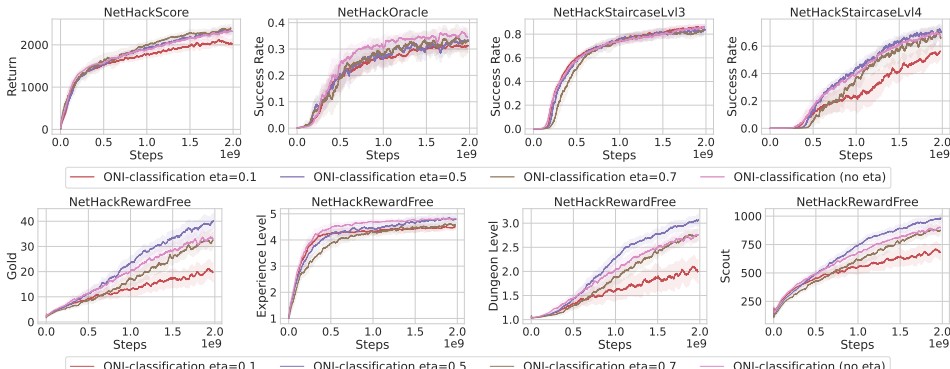

**Figure 5.3:** **(Top)** Performance of `ONI-classification` is robust to different choices of the classification threshold $\eta$ on the four NetHack tasks. **(Bottom)** $\eta = 0.5$ marginally outperform the other values for training intrinsic-reward-only agents. The result when using $P(y_t = 1|o_t)$ as reward (6) is marked with legend `oni-classification (no eta)`.

**Performance vs. LLM Annotation Throughput** We compared agents trained on the `Score` and `Oracle` tasks using either 1 or 4 Tesla V100-32GB GPUs in the LLM server node. As shown in Figure 5.4, using 4 GPUs rather than 1 significantly increases the number of annotated observations. However, we do not find any significant change in performance between the two. This suggests that

---

[7]To increase the diversity of the captions in the training dataset, we do not requery the LLM server if a caption is already annotated before. Therefore, every caption in our dataset only has a single label, resulting in this phenomenon.

many of the labelled examples may contain redundant information, which is not useful for updating the reward model. Designing more sophisticated prioritization schemes, which can select maximally informative examples to send to the LLM, constitutes an interesting direction for future work.

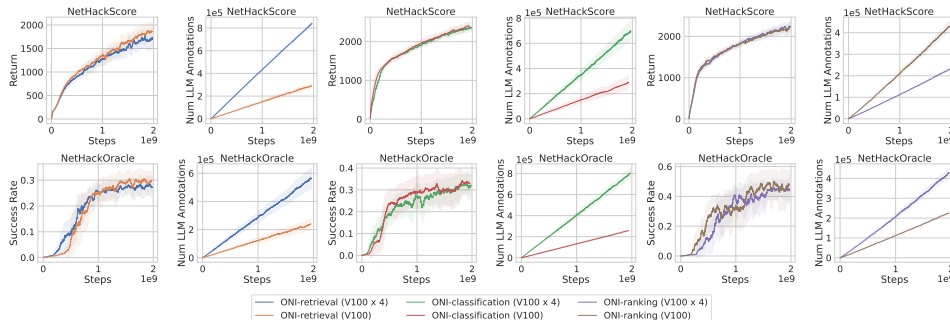

**Figure 5.4:** Performance remains comparable despite doubling LLM annotations, as seen in results with LLM server utilizing 1 GPU vs. 4 GPUs. The number of annotations received by `ONI-ranking` is lower than the other two, as the LLM server analyzes two messages for each annotation (see the prompts in Appendix B).

**Performance When Reducing LLM Annotations** Thus far we have been using all LLM annotations available. It is also intriguing to check the performance when we limit the volume of annotations, to simulate more resource-constrained settings. Here we subsample the LLM-annotated messages with rate 0.1 and 0.01 before sending them back to the hash table. Figure 5.5 shows that the performance of `ONI-retrieval` significantly drops with rate 0.01, whereas `ONI-classification` remains comparable. This suggests that using a parametric reward model can help reduce the number of annotations required thanks to its generalization ability.

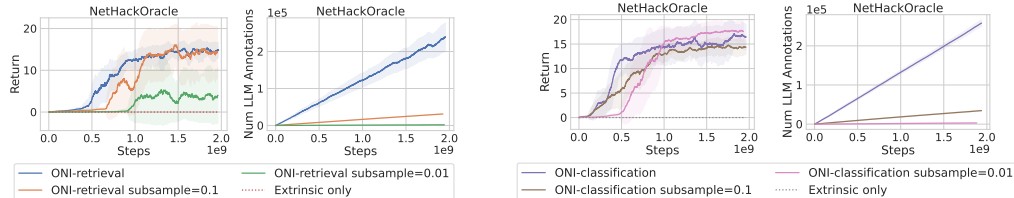

**Figure 5.5:** Performance of `ONI-retrieval` plunges when the subsampling rate reduces to 0.01, while performance of `ONI-classification` is still comparable with the original one.

**Impact of Intrinsic Reward Coefficient** $\beta$ For `ONI-classification` and `ONI-retrieval`, we have used different values of $\beta$ for the `Score` task and other sparse reward tasks in Section 5.1. Throughout our experiments, we have found that for these two methods, larger values of $\beta$ lead to better performance for the sparse reward tasks, while smaller values of $\beta$ better balance between intrinsic and extrinsic rewards for the dense reward `Score` task. In comparison, `ONI-ranking` is relatively robust to this choice, where larger values of $\beta$ are still slightly favored for the sparse reward tasks. We include details in Appendix C.3.

## 6 CONCLUSION

We have introduced `ONI`, a distributed online intrinsic reward and agent learning system. We showed that we are able to match the state of the art across a range of challenging sparse reward tasks from the NetHack Learning Environment, while removing the need for a large pre-collected dataset or auxiliary dense reward function required by previous work. We explored three different instantiations of our system of varying levels of complexity and generality, and study their tradeoffs. Our work paves the way for intrinsic reward methods which can learn purely from agent experience, are not constrained by external dataset size or quality, and can leverage high-performance RL training.

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

## A EXPERIMENTAL DETAILS

### A.1 ARCHITECTURES

Our architectures largely follow those used in Klissarov et al. (2023).

**Our policy network** uses the Chaotic Dwarven GPT5 architecture originally introduced in Myf-fili (2021). This architecture combines convolutional layers processing the top-down visible map centered at the agent with fully-connected layers processing messages and bottom-line statistics including hit points, experience, hunger level and the like. The convolutional encoder has 3 convolutional layers with $32, 64, 128$ feature maps respectively, interleaved with exponential linear unit (ELU) non-linearities (Clevert et al., 2016). Messages and bottom-line statistics are each processed with 2-layer MLPs with 128 hidden units each. All embeddings are combined, passed through a single-layer MLP with 512 hidden units, and then passed to a recurrent GRU module (Cho et al., 2014) with 512 hidden units. Finally, this hidden representation feeds into linear critic and actor heads.

**Our reward model** The reward model of `ONI-ranking` is based on the encoder from Küttler et al. (2020) that processes both state representation and messages. Messages are processed by a 5-layer character-level CNN (Zhang et al., 2015) with 64 feature maps at each layer. The first, second and last layers are interleaved with max-pooling layers with kernel size and stride 3. The output is then passed through a 3-layer MLP with $128, 256, 512$ hidden units at each layer respectively, and ReLU non-linearities, followed by a scalar output. The reward model of `ONI-classification` only process messages, using the same architecture described above.

### A.2 HYPERPARAMETERS

Following Klissarov et al. (2023), we scale the environment reward by 0.1 for the `Score` task and by 10 for the other sparse reward tasks, and use normalized intrinsic reward

$$r^{\text{int}}_{\text{normalized}}(o_t) = r^{\text{int}}(o_t)/N(c_t)^z, \tag{7}$$

where $N(c_t)$ is the number of times the caption $c_t$ has been found in one episode. For all our experiments, we use $z = 3.0$. This is also called the *episodic bonus* (Henaff et al., 2022b).

For `ONI-classification` and `ONI-ranking`, we train the reward model using the Adam optimizer (Kingma, 2014) with batch size 256. `ONI-classification` is trained with learning rate 0.0001, classification threshold $\eta = 0.7$, $\beta = 0.1$ for the `Score` task and $\beta = 0.4$ for the others. `ONI-ranking` is trained with 0.00001, $\beta = 0.05$ and $\nu_{\mathcal{N}} = 1.96$ (97.5-th quantile of the standard normal distribution). `ONI-retrieval` does not train a reward model, and we use $\beta = 0.1$ for the `Score` task and $\beta = 0.5$ for the others.

Table A.1 shows the APPO hyperparameters which are common to all experiments and Table A.2 includes the hyperparameters for the online LLM annotation.

| Hyperparameter | Value |
|---|---|
| Number of Parallel Environment Instances | 480 |
| Batch Size | 4096 |
| PPO Clip Ratio | 0.1 |
| PPO Clip Value | 1.0 |
| PPO Epochs | 1 |
| Max Gradient Norm | 4.0 |
| Value Loss Coefficient | 0.5 |
| Exploration Loss | entropy |

**Table A.1:** Common APPO hyperparameters across all experiments.

| Hyperparameter | Value |
|---|---|
| LLM Model | LLaMA-3.1-8B-Instruct |
| Temperature | 0.1 |
| Max tokens to generate | 4096 |
| Annotation Batch Size | 100 |

**Table A.2:** LLM annotation hyperparameters.

### A.3 IMPLEMENTATION DETAILS FOR ELLM-BoW

For any piece of text $c$, we construct an embedding vector by first applying the FastText tokenizer (Bojanowski et al., 2016) to the text, separating it into words $c = (w_1, ..., w_m)$. We then retrieve the FastText word embeddings $x_1, ..., x_m$ for each word, and compute their sum $\texttt{embed}(c) = \sum_{i=1}^{m} x_i$. The intrinsic reward is then defined by the cosine distance between the embeddings of the caption of the current observation and the goal, weighted by an episodic count-based term::

$$r(o_t) = \frac{1}{N_e(c_t)^z} \cdot \frac{\texttt{embed}(c_t) \cdot \texttt{embed}(g)}{\|\texttt{embed}(c_t)\| \cdot \|\texttt{embed}(g)\|}$$

We include the episodic term for consistency with Motif and ONI. In Section C.7, we ablate this choice and show that it is essential for good performance.

## B  PROMPTS

Here we include all the LLM prompts we use. The system prompt and the prompt for ONI-ranking are identical to the ones used in Klissarov et al. (2023). We adapted these to produce labels for single observations for ONI-retrieval and ONI-classification.

**System Prompt (used by all methods)**

```
You are a helpful and honest judge of good gameplaying and progress in
the NetHack game. Always answer as helpfully as possible, while being
truthful.

If you don't know the answer to a question, please don't share false
information.
```

**Prompt for ONI-retrieval and ONI-classification**

```
Context: I am playing NetHack and I will present you a message from the
gameplay.

Instruction: First, tell me about your knowledge of NetHack. Mention the
goal of NetHack.
Then, write an analysis describing the semantics of each message using
your knowledge of NetHack. Finally, respond by assigning a score to the
message, which measures whether the status represented by the message is
helpful or unhelpful to make some progress towards the goal.
Helpful here does not mean whether it is informative to the player to
make decisions or not, but whether it is helpful for the agent to solve
the game.
I prefer agents that maximize the score in the game, for instance by
killing monsters, collecting gold or going down the stairs in the dungeon
.
For extremely helpful messages, assign a label FOO. For unhelpful,
neutral, and lukewarm messages, assign a label BAR.

Example 1: "You kill the goblin!  Welcome to experience level 4"
indicates that the player
has killed a goblin and gained experience. This led to a level up and
increased abilities.
Gaining experience and leveling up are both extremely helpful and this
message should be labelled as FOO.

Example 2: "You hear someone counting money." indicates that the player
is close to a source of gold, as the sound of counting money suggests a
nearby NPC or monster with treasure.
This is also extremely helpful and should be labelled as FOO.

Example 3: "5 gold pieces." indicates you collected 5 gold coins. This is
 extremely helpful should be labelled as FOO.

Example 4: "That door is closed." is neutral and should be labelled as
BAR.

Please rate this message: {{"{}"}}

Constraint: Please format your response strictly in this format without
additional explanation:
<knowledge> [your knowledge of NetHack] </knowledge>
<analysis> [your one-sentence analysis of the message] </analysis>
<label> [FOO/BAR] </label>
```

**Prompt for `ONI-ranking`**

```
I will present you with two short gameplay descriptions.
First, tell me about your knowledge of NetHack. Mention the goal of
NetHack. Prefer agents that maximize the score in the game, for instance
by killing monsters, collecting gold or going down the stairs in the
dungeon.
Then, write an analysis describing the semantics of each description
strictly using information from the descriptions (which may be empty) and
 your knowledge of NetHack.
Provide a comparative analysis based on first princicples.
Finally, respond by explicitly declaring which one is the most likely to
make some progress towards the goal, writing either ("best_description":
1), ("best_description": 2). You could also say ("best_description": None
).

{{
"description_1":
"{}"
}}

{{
"description_2":
"{}"
}}
```

# C ADDITIONAL RESULTS

## C.1 LLAMA-3.1-70B-INSTRUCT VS LLAMA-3.1-8B-INSTRUCT

In Figure C.1, we compare the performance of Motif using two different sized LLMs on `Score` and `Oracle` (LLaMA-3.1-8B-Instruct and LLaMA-3.1-70B-Instruct). Interestingly, we do not observe a significant difference between the two. This is in contrast to the previous work of (Klissarov et al., 2023), who found a significant difference between using LLaMA-2-70B-chat and LLaMA-2-7B-chat. This suggest that the smaller 8B model is sufficient for evaluating messages on these tasks, hence we use it in our experiments.

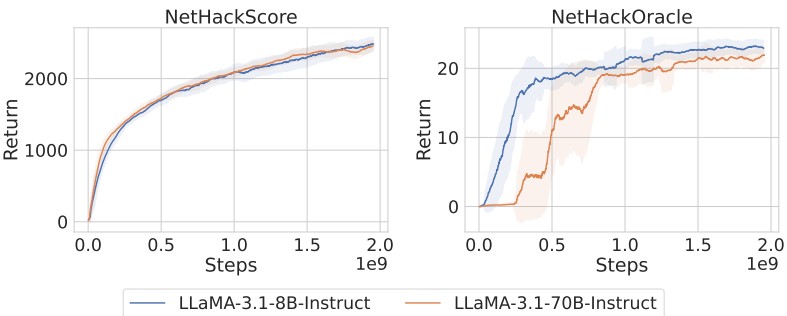

**Figure C.1:** Performance of Motif using two different LLMs. Curves represent means and shaded regions represent standard errors over 5 seeds.

## C.2 PERFORMANCE VS. LLM THROUGHPUT: TESLA V100 VS TESLA A100 GPU

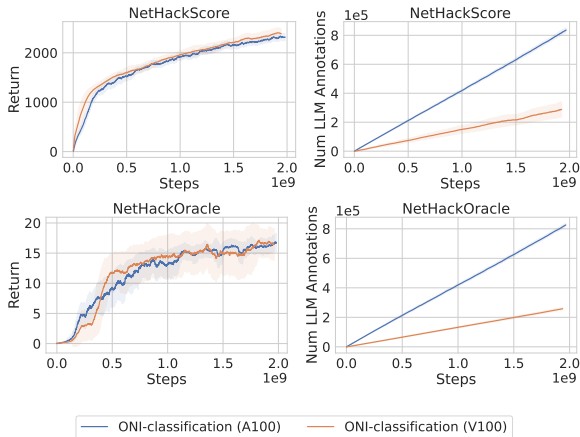

**Figure C.2:** Performance of `ONI-classification` is comparable when the LLM server uses a Tesla A100-80GB or V10-32GB GPU.

## C.3 IMPACT OF INTRINSIC REWARD COEFFICIENT $\beta$

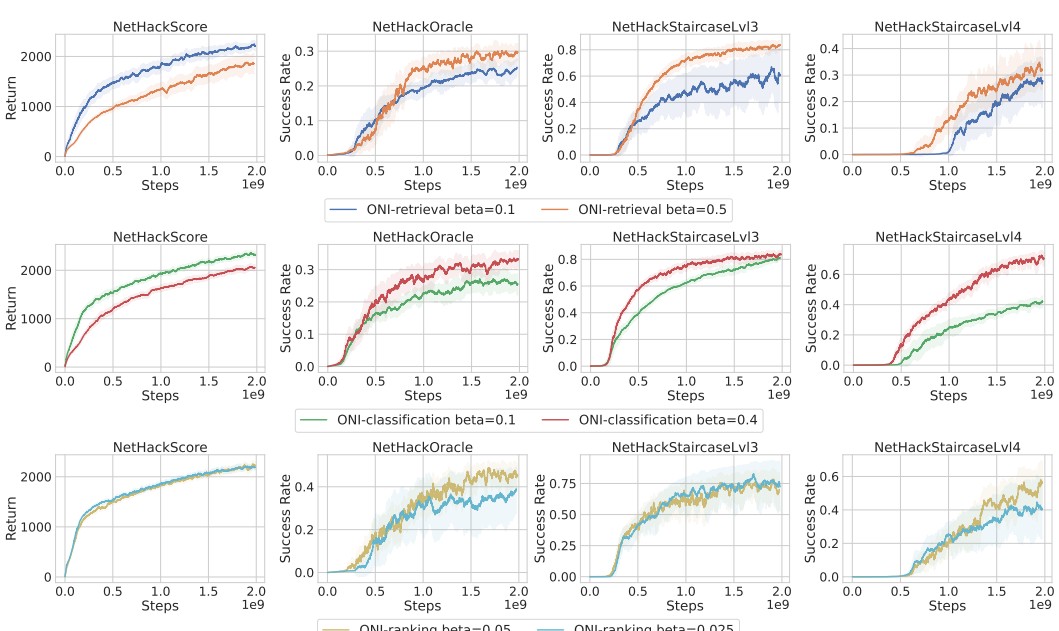

**Figure C.3:** For `ONI-retrieval` and `ONI-classification`, sparse reward tasks favor larger intrinsic reward coefficient $\beta$ while smaller values of $\beta$ lead to better results for the dense reward `Score` task. For `ONI-ranking`, we do not observe much difference for the `Score` task, but the sparse reward tasks still slightly favors larger $\beta$.

## C.4 PERFORMANCE WHEN REDUCING LLM ANNOTATIONS

To investigate the effect of reducing the number of LLM annotations, we subsample the LLM annotated messages before sending them back to the hash table (see Section 3.2). Figure 5.5 shows the performance of `ONI-retrieval` and `ONI-classification` on `Oracle` when the subsampling rate is 0.1 and 0.01. `ONI-classification` is more robust than `ONI-retrieval`, where its performance is still on par with the original one even when the subsampling rate reduces to 0.01, whereas `ONI-retrieval`'s performance drops significantly.

## C.5 ADDITIONAL ABLATIONS

**`ONI-ranking`: the impact of sampling strategy for LLM annotation and reward training** Unlike `ONI-classification` that uses a LIFO queue to rank captions for LLM annotation, it is more subtle to design the most effective sampling strategy for `ONI-ranking`, where we need to construct pairs of captions to annotate and use for reward model training. Here we study the effect of deduplicating captions before passing them to our pipeline. In either case, we maintain a message list $\mathcal{L}$, and sample pairs of captions $c_1, c_2 \sim \text{Uniform}(\mathcal{L} \times \mathcal{L})$ for annotation. The annotated message pairs are stored in another list, which is sampled from uniformly when training the reward model. In the first option, each time the agent encounters a message, we check if it is already stored in $\mathcal{L}$ and only add it if not. This is similar to the approach used in `ONI-classification`. In the second option, we simply add all captions encountered by the agent into $\mathcal{L}$, regardless of whether they have already been seen before. This approach is similar to that taken by the original Motif work, which does not perform any deduplication of the offline dataset.

Figure C.4 shows the performance and the number of captions stored in the replay buffer for both schemes. We see that for the deduplicated variant, the number of captions does not grow past a certain point (approximately $70k$ unique captions). For the non-deduplicated variant, the number of captions keeps growing linearly over time. The deduplicated variant fails to learn, which highlights the important effect which the annotation dataset can have. We hypothesize that the deduplicated variant may undersample captions that occur frequently in the agent's experience, for which it is im-

portant to reliably estimate reward. For example, the blank message occurs very frequently during policy learning, but is only included in a small fraction of the pairs sent to the LLM for annotation, since it is sampled with the same probability as the other $\sim 70k$ other captions. In contrast, the duplicated variant samples the blank message and other frequent captions with much higher probability. Still, this remains a simple strategy, and designing more sophisticated sampling mechanisms (for example, that account for the epistemic uncertainty of the reward model) would be an interesting direction for future work.

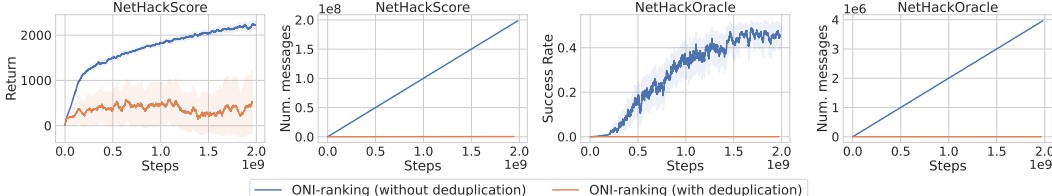

**Figure C.4:** Performance and number of captions in the replay buffer for agents trained with and without message deduplication. Keeping the natural distribution of captions, and not deduplicating, is important to enable our ranking-based method to learn.

## C.6 GOAL STRING AND PROMPTS USED IN SECTION 5.2

For the experiments in Section 5.2, we use the goal strings from (Klissarov et al., 2023) used to induce different behaviors in agents, such as the Gold Collector, Monster Slayer or Descender. These goal string explicitly aim to maximize one metric while minimize the others. The default goal string is the same used in `ONI` prompts.

### C.6.1 PROMPT USED FOR `ELLM-BoW`

**Default Goal**

```
Prefer agents that maximize the score in the game, for instance by
killing monsters, collecting gold or going down the stairs in the dungeon
.
```

**Combat Goal**

```
Prefer agents that engage in combat, for instance by killing monsters.
But never prefer agents that collect ANY gold or that go down the dungeon
.
```

**Gold Goal**

```
Prefer agents that maximize their gold. But never prefer agents that
maximize the score in other ways (e.g., by engaging in combat or killing
monsters) or that go down the dungeon.
```

### C.6.2 PROMPT USED FOR `ONI-RETRIEVAL`

The prompt used for `ONI` methods have been modified to reflect the change of goals, which we list in the next subsection. In addition to the changes of goal description, we use the same examples as in Appendix B but changed its label.

**Combat Goal**

```
Context: I am playing NetHack and I will present you a message from the
gameplay.

Instruction: First, tell me about your knowledge of NetHack. Mention the
goal of NetHack.
Then, write an analysis describing the semantics of each message using
your knowledge of NetHack. Finally, respond by assigning a score to the
```

```
message, which measures whether the status represented by the message is
helpful or unhelpful to make some progress towards the goal.
Helpful here does not mean whether it is informative to the player to
make decisions or not, but whether it is helpful for the agent to solve
the game.
I prefer agents engage in combat, for instance by killing monsters. But
never prefer agents that collect ANY gold or that go down the dungeon.
For extremely helpful messages, assign a label FOO. For unhelpful,
neutral, and lukewarm messages, assign a label BAR.

Example 1: "You kill the goblin!  Welcome to experience level 4"
indicates that the player
has killed a goblin and gained experience. This led to a level up and
increased abilities.
Gaining experience and leveling up are both extremely helpful and this
message should be labelled as FOO.

Example 2: "You hear someone counting money." indicates that the player
is close to a source of gold, as the sound of counting money suggests a
nearby NPC or monster with treasure.
This is not helpful and should be labelled as BAR.

Example 3: "5 gold pieces." indicates you collected 5 gold coins. This is
 not aligned with our goal and should be labelled as BAR.

Example 4: "That door is closed." is neutral and should be labelled as
BAR.

Please rate this message: {{"{}"}}

Constraint: Please format your response strictly in this format without
additional explanation:
<knowledge> [your knowledge of NetHack] </knowledge>
<analysis> [your one-sentence analysis of the message] </analysis>
<label> [FOO/BAR] </label>
```

**Gold Goal**

```
Context: I am playing NetHack and I will present you a message from the
gameplay.

Instruction: First, tell me about your knowledge of NetHack. Mention the
goal of NetHack.
Then, write an analysis describing the semantics of each message using
your knowledge of NetHack. Finally, respond by assigning a score to the
message, which measures whether the status represented by the message is
helpful or unhelpful to make some progress towards the goal.
Helpful here does not mean whether it is informative to the player to
make decisions or not, but whether it is helpful for the agent to solve
the game.
I prefer agents that maximize their gold. But never prefer agents that
maximize the score in other ways (e.g., by engaging in combat or killing
monsters) or that go down the dungeon.
For extremely helpful messages, assign a label FOO. For unhelpful,
neutral, and lukewarm messages, assign a label BAR.

Example 1: "You kill the goblin!  Welcome to experience level 4"
indicates that the player
has killed a goblin and gained experience.  This is not aligned with our
goal and should be labelled as BAR.

Example 2: "You hear someone counting money." indicates that the player
is close to a source of gold, as the sound of counting money suggests a
nearby NPC or monster with treasure. This is extremely helpful and should
 be labelled as FOO.
```

```
Example 3: "5 gold pieces." indicates you collected 5 gold coins. This is
 extremely helpful and should be labelled as FOO.

Example 4: "That door is closed." is neutral and should be labelled as
BAR.

Please rate this message: {{"{}"}}

Constraint: Please format your response strictly in this format without
additional explanation:
<knowledge> [your knowledge of NetHack] </knowledge>
<analysis> [your one-sentence analysis of the message] </analysis>
<label> [FOO/BAR] </label>
```

### C.7    EFFECT OF THE EPISODIC TERM FOR ELLM-BOW

Our implementation of `ELLM-BoW` has included the episodic-count based normalization (7), which is key to the performance of `ELLM-BoW`. Figure C.5 shows that directly using ELLM-BoW as in Du et al. (2023a) `ELLM-BoW`, without the episodic term, failed to make progress in all three tasks.

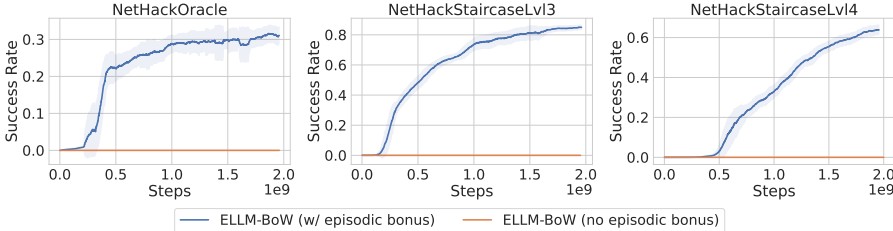

**Figure C.5:** `ELLM-BOW` performs well when the intrinsic reward is normalized by an episodic-count based term as in (7). Without it, the success rate is zero for all the three tasks.

## D    ADDITIONAL RELATED WORK

**LLM for RL Broadly**  Another way of leveraging the prior knowledge encoded in LLMs for decision making is to use the LLM directly as a policy. This approach has been successfully used in robotics (Ahn et al., 2022; Driess et al., 2023), as well as open-ended exploration in MineCraft (Wang et al., 2024). Both settings require the LLM to operate at a higher level of abstraction, by having it call upon a set of semantically grounded skills which handle the low-level sensorimotor activity. These are in turn produced by imitation learning on expert trajectories or hardcoded APIs. Jeurissen et al. (2024) prompt the LLM to choose a predefined skill to play NetHack. The prompts are constructed to represent past events, current observation, and the task description and hardcoded available skills are also included. More references on LLMs for decision making can be found in the survey paper of Cao et al. (2024).

