# OpenReview forum: "Online Intrinsic Rewards for Decision Making Agents from Large Language Model Feedback"
_ICLR.cc/2025/Conference — Submitted to ICLR 2025_

### Official Review · Reviewer_C2KC · 2024-10-31

**Soundness:** 2
**Presentation:** 3
**Contribution:** 2
**Rating:** 5
**Confidence:** 4

**Summary:**

ONI introduces a novel distributed framework for developing intrinsic rewards in reinforcement learning by leveraging feedback from large language models (LLMs). This approach supports policy optimization in sparse reward scenarios without needing external datasets. By exploring several methods for reward modeling, ONI demonstrates impressive performance in the NetHack environment, closely matching the results of previous approaches like Motif while eliminating the requirement for pre-collected data

**Strengths:**

- ONI provides a practical and innovative way to use feedback from large language models as intrinsic rewards, making it especially valuable for reinforcement learning environments where extrinsic rewards are sparse.

- By eliminating the need for external datasets and instead building rewards from the agent’s own experience, ONI offers a scalable approach that could adapt to a wide range of tasks.

- The asynchronous design keeps the policy training process uninterrupted, maintaining high throughput and efficiency throughout.

- ONI shows strong, competitive performance in challenging tasks within the NetHack environment, matching or even surpassing state-of-the-art methods in tasks like Oracle and multi-level exploration without relying on dense reward functions.

- Enough details are provided to reproduce the method. The method reproducibility appears to be good.

**Weaknesses:**

- The method is primarily evaluated in the NetHack environment, leaving questions on how well ONI would generalize to other, potentially more diverse RL environments. It would be critical to show that the method can be applied to other environments and domains. In the current state, evaluation is too narrow to demonstrate the impact of this work.

- Asynchronous LLM feedback, while effective here, may face scalability challenges in more complex settings where feedback needs scale up significantly. The paper lacks detailed discussion on this issue.

- The paper should discuss and report evaluation with different LLM. In the current state, the results cannot demonstrate that the method can be used with other LLMs.

- LLM-generated annotations may carry inherent biases that can influence reward generation. The paper lacks a strategy for identifying or mitigating such biases.

- Comparisons with other intrinsic reward models and exploration techniques are minimal, limiting clarity on ONI's unique advantages or limitations relative to a wider range of baselines. I would suggest for instance comparing current paper with strong LLM-guided baselines, such as [“Guiding Pretraining in Reinforcement Learning with Large Language Models”, Du et al., 2023], [“Exploring Beyond Curiosity Rewards: Language-Driven Exploration in RL”, Bougie et al., 2024], or other recent papers.

**Questions:**

- How would ONI perform in other environments beyond NetHack, especially ones with more complex or variable observation spaces?

- Are there any insights into how specific LLM choices impact ONI's performance, especially for larger models?

- Could potential biases in LLM annotations affect the intrinsic reward generation process, and if so, how are they mitigated?

- How does the method compare with stronger recent baselines that guide exploration with LLMs?

---

> ### Author Response · Authors · 2024-11-26
>
> Thank you for the constructive review. We give detailed answers to your questions below, please also see our common response above.
>
> > The method is primarily evaluated in the NetHack environment…How would ONI perform in other environments beyond NetHack, especially ones with more complex or variable observation spaces?
>
> Using LLMs currently limits us to domains that have textual observations, and among these, we believe Nethack to be among the most challenging. This is corroborated by the recently introduced BALROG benchmark, which shows that among 6 text-based benchmarks, Nethack is the one where SOTA LLMs do the most poorly (https://balrogai.com/).
>
> In principle, the LLM could be replaced by a multimodal VLM to process visual observations. At the time of submission, there were no high-performing open-source multi-modal LLMs, hence we did not investigate this. Since then, high-quality VLMs have become available (e.g. LLaMA 3.1-V and Pixtral), and we are currently investigating them.
>
>
> > Asynchronous LLM feedback, while effective here, may face scalability challenges in more complex settings where feedback needs scale up significantly. The paper lacks detailed discussion on this issue.
>
> Depending on the problem and hardware available, our framework supports different setups. On one hand, we can easily increase the LLM throughput by increasing the number of GPUs and nodes if they are available. In this case, the main bottleneck becomes communication over the network. We can also use the GPUs on the same node as the GPU used to train the agent, in which case communication costs become minimal (but the LLM throughput is limited by the number of GPUs on the node). However, the communication cost is inevitable. We have added clarification on this point in Section 5.1.
>
> > The paper should discuss and report evaluation with different LLM. In the current state, the results cannot demonstrate that the method can be used with other LLMs.
>
> In Appendix C.1 we compare LLaMA-3.1-8B-Instruct and LLaMA-3.1-70B-Instruct for offline Motif, and find that they are similar. In early experiments, we also found that these two LLMs performed similarly in the online setting. We will rerun these experiments with our latest setup and include results in the camera ready, if the paper is accepted.
>
>
>
> > LLM-generated annotations may carry inherent biases that can influence reward generation. The paper lacks a strategy for identifying or mitigating such biases…Could potential biases in LLM annotations affect the intrinsic reward generation process, and if so, how are they mitigated?
>
> This is true, but we believe this affects all methods for generating intrinsic rewards using LLMs and is a separate (and interesting) research question. Our focus here is specifically on scaling up to high-throughput settings and removing the dependence on offline datasets. Finding ways to robustify these kinds of methods to blind spots in the LLM would be interesting future work.
>
> > Comparisons with other intrinsic reward models and exploration techniques are minimal, limiting clarity on ONI's unique advantages or limitations relative to a wider range of baselines…How does the method compare with stronger recent baselines that guide exploration with LLMs?
>
> This is a good point, and we have added such a comparison to the updated paper. Our setting where high throughput is required limits the methods we can run, because many LLM-based methods for intrinsic reward generation require an LLM call for each agent observation. We adapted the suggested method of [1], please see our more detailed discussion in the common response above and the updated paper.
>
> We hope that we have addressed your concerns, and that you will consider increasing your score as a result. If you have any remaining concerns, please let us know and we will be happy to discuss them.
>
> [1]  “Guiding Pretraining in Reinforcement Learning with Large Language Models”, Du et al., 2023

---

> ### Author Response · Authors · 2024-12-02
>
> Dear reviewer,
>
> Thank you again for your time reviewing our paper. Given that the discussion period ends tomorrow, could you kindly let us know if our response above adequately answers the questions/concerns you had in your review, and if so, update your review? If not, please also let us know and we will do our best to address any remaining concerns.

---

### Official Review · Reviewer_Cv7N · 2024-11-03

**Soundness:** 2
**Presentation:** 3
**Contribution:** 1
**Rating:** 3
**Confidence:** 4

**Summary:**

This paper introduces ONI, a distributed architecture that simultaneously learns a RL policy and an intrinsic reward function using LLM feedback. ONI is built upon Sample Factory, an asynchronous reinforcement learning framework, and incorporates an LLM server hosted on a separate node to annotate observations, facilitating the learning of an intrinsic reward model that improves the sample efficiency of RL algorithms in sparse reward environments. This paper evaluates several designs of the intrinsic reward function and demonstrates ONI’s strong performance in the NetHack Learning Environment.

**Strengths:**

- The paper is well-written, and the method is clearly introduced.
- The limitations of previous approaches are effectively summarized, and the problem investigated is significant.

**Weaknesses:**

- My primary concern lies in the novelty of the contribution. The proposed method, ONI, appears heavily intertwined with Sample Factory[1], adding only a minor extension. The distributed architecture should, in my view, be credited to Sample Factory rather than this work. Furthermore, both the LLM-based annotation and the intrinsic reward design have been previously introduced in existing research[2,3]. As a result, I believe this paper may not be suitable for acceptance at top-tier venues.
- Can the assumption of access to observation captions be relaxed, perhaps by leveraging a multi-modal LLM?
- The experimental domain is limited to the NetHack Learning Environment. Could the method be extended to other widely-used RL benchmarks, such as MuJoCo[4] or Atari[5]?
- The authors noted that increasing LLM annotations did not significantly improve performance. Could computational resources be optimized by reducing the number of annotations? What is the minimum level of annotation needed to maintain performance?
- The discussion on the method's limitations remains insufficient.
- The code is not available.

[1] Aleksei Petrenko, Zhehui Huang, Tushar Kumar, Gaurav S. Sukhatme, and Vladlen Koltun. Sample factory: Egocentric 3d control from pixels at 100000 FPS with asynchronous reinforcement learning. In Proceedings of the 37th International Conference on Machine Learning, ICML 2020, 13-18 July 2020, Virtual Event, volume 119 of Proceedings of Machine Learning Research, pp. 7652–7662. PMLR, 2020. URL http://proceedings.mlr.press/v119/ petrenko20a.html.

[2] Lee H, Phatale S, Mansoor H, et al. RLAIF vs. RLHF: Scaling Reinforcement Learning from Human Feedback with AI Feedback[C]//Forty-first International Conference on Machine Learning.

[3] Martin Klissarov, Pierluca D’Oro, Shagun Sodhani, Roberta Raileanu, Pierre-Luc Bacon, Pascal Vincent, Amy Zhang, and Mikael Henaff. Motif: Intrinsic motivation from artificial intelligence feedback. arXiv preprint arXiv:2310.00166, 9 2023.

[4] Todorov E, Erez T, Tassa Y. Mujoco: A physics engine for model-based control[C]//2012 IEEE/RSJ international conference on intelligent robots and systems. IEEE, 2012: 5026-5033.

[5] Bellemare M G, Naddaf Y, Veness J, et al. The arcade learning environment: An evaluation platform for general agents[J]. Journal of Artificial Intelligence Research, 2013, 47: 253-279.

**Questions:**

Please take a look at the Weaknesses section.

---

> ### Author Response · Authors · 2024-11-26
>
> Thank you for the feedback. We are glad that you found our paper well-written, the problem we aim to address is significant, and that our method demonstrates strong performance. We answer your concerns in detail below.
>
> > My primary concern lies in the novelty of the contribution…
>
> We respectfully disagree with the claim that our work is not sufficiently novel. Our work enables novel capabilities: it allows online training RL agents at scale using intrinsic rewards, without requiring large precollected datasets. As discussed in our related work section, prior work either: i) uses LLMs to label *all* data collected online, which quickly runs into computational bottlenecks, or ii) requires large precollected data to first synthesize the intrinsic reward function, which limits applicability.
>
>
>
>
> > Can the assumption of access to observation captions be relaxed, perhaps by leveraging a multi-modal LLM?...The experimental domain is limited to the NetHack Learning Environment. Could the method be extended to other widely-used RL benchmarks, such as MuJoCo[4] or Atari
>
> Using LLMs currently limits us to domains that have textual observations, and among these, we believe Nethack to be among the most challenging. This is corroborated by the recently introduced BALROG benchmark, which shows that among 6 text-based benchmarks, Nethack is the one where SOTA LLMs do the most poorly (https://balrogai.com/).
>
> In principle, the LLM could be replaced by a multimodal VLM to process visual observations. At the time of submission, there were no high-performing open-source multi-modal LLMs, hence we did not investigate this. Since then, high-quality VLMs have become available (e.g. LLaMA 3.1-V and Pixtral), and we are currently investigating them.
>
> > The authors noted that increasing LLM annotations did not significantly improve performance. Could computational resources be optimized by reducing the number of annotations? What is the minimum level of annotation needed to maintain performance?
>
> Thank you for the suggestion, we have added additional experiments investigating this. We found that the retrieval-based method degrades fairly quickly as we reduce the number of annotations, but the classification-based method is a lot more robust. See Figure 5.5 in the updated version. This highlights one of the advantages of using a reward model rather than simple caching.
>
>
> > The code is not available.
>
> Our abstract explicitly states that our code will be open sourced. The URL included there is a placeholder. We will share the code upon publication.
>
> We hope that our answers and additional experiments have addressed your concerns, and that you will consider raising your score to Accept. If there are still remaining concerns that prevent this, please let us know as soon as possible and we will be happy to discuss further.

---

> ### Author Response · Authors · 2024-12-02
>
> Dear reviewer,
>
> Thank you again for your time reviewing our paper. Given that the discussion period ends tomorrow, could you kindly let us know if our response above adequately answers the questions/concerns you had in your review, and if so, update your review? If not, please also let us know and we will do our best to address any remaining concerns.

---

### Official Review · Reviewer_9m6i · 2024-11-03

**Soundness:** 3
**Presentation:** 2
**Contribution:** 2
**Rating:** 5
**Confidence:** 3

**Summary:**

The paper proposes a method to simultaneously learn an RL policy and intrinsic reward function using feedback from LLMs, which annotate the agent’s experiences. The method is shown to outperform competing approaches in the NetHack environment.

**Strengths:**

ONI seems to be a novel approach with promising results for learning intrinsic rewards. Leveraging LLMs to learn or infer reward functions is an interesting and relevant contemporary topic.

**Weaknesses:**

In general, the writing could be improved for clarity. There are some sections with unnecessarily long sentences, which impeded understanding. The work could be scoped out better to clarify exactly what the contributions are.

**Questions:**

1.	What is the impact of limited annotation on the reward model? Also what is the impact of the quality of annotations?
2.	The lack of difference in performance between ranking and classification methods could be investigated further. Analyzing the distribution of intrinsic rewards generated by each method or comparing their performance on tasks with higher observation diversity could offer further insights. Have the authors considered this?
3.	It would be good to scope out clearly what the method is trying to achieve. For example, is the intrinsic reward used during learning fully learnt or is it just a current estimate?
4.	Is there a difference in the final reward model learnt via ONI and one learnt using say, a good amount of offline data? Does the reward from ONI converge? I wonder whether theoretical properties relating to this have been studied.
5.	The logic behind eq 6 should be clearly and explicitly mentioned immediately after the equation.
6.	Although it may not be directly relevant, it would be reassuring to include additional baselines (or modified versions of it) based on the reported related work.
7.	What does ONI stand for?
8.	The paper has some unusually long sentences (such as in the abstract), which should be split up for simplicity and clarity.
9.	In section 3, it would have been better to focus on the proposed approach rather than elaborating on the details in say, lines 142-148 for example. These details are not central to the understanding of ONI and could have been pushed to the appendix.
10.	URL in the abstract does not work
11.	$\eta$ is not rendered properly in Fig 5.2

---

> ### Author Response · Authors · 2024-11-26
>
> Thank you for the review and suggestions for improvement. We answer your questions below:
>
>
> > In general, the writing could be improved for clarity. There are some sections with unnecessarily long sentences, which impeded understanding…The paper has some unusually long sentences (such as in the abstract), which should be split up for simplicity and clarity.
>
> We have simplified the longer sentence in the abstract we think you are referring to. Please let us know if there are other parts you find unclear and we will do our best to improve them. It would be helpful to know which other parts you are referring to, if there are any.
>
>
> > The work could be scoped out better to clarify exactly what the contributions are. It would be good to scope out clearly what the method is trying to achieve.
>
> The contributions and scope are: current methods either do not scale to high-throughput settings, because they require LLM annotation of the data collected online to compute rewards, or they require large offline datasets with which to first learn a reward function. Our method is both scalable, and does not require any offline data, which may not exist or be difficult to collect. We hope our updated abstract makes this clear, let us know if not and we will further update the paper.
>
>
> > For example, is the intrinsic reward used during learning fully learnt or is it just a current estimate?
>
> The intrinsic reward is learned at the same time as the policy, while the data is being annotated. At each update, the intrinsic reward is computed using the current state of the reward model, which changes over time as more data is collected and annotated.
>
> > What is the impact of limited annotation on the reward model?
>
> This is a good suggestion, we have added an additional ablation studying this in Section 5.3 (Figure 5.5). We also discuss this in our common response at the top.
>
> > Is there a difference in the final reward model learnt via ONI and one learnt using say, a good amount of offline data? Does the reward from ONI converge? I wonder whether theoretical properties relating to this have been studied.
>
> The “reward model learned using a good amount of offline data” is exactly the Motif method we compare to. Its performance tends to be slightly higher, but it requires the stronger assumption that a diverse offline dataset is available. Running Motif also involves a more laborious 3-stage process of running LLM annotations, training a reward model, and finally training the policy, whereas ONI operates as a single process.
>
> > The logic behind eq 6 should be clearly and explicitly mentioned immediately after the equation.
>
> We have added a sentence providing more clarification.
>
> > Although it may not be directly relevant, it would be reassuring to include additional baselines (or modified versions of it) based on the reported related work.
>
> We have done so, please see our common response to all reviewers above.
>
> > What does ONI stand for?
>
> It stands for ONline Intrinsic Rewards, we will clarify this in the update paper and code release.
>
> > In section 3, it would have been better to focus on the proposed approach rather than elaborating on the details in say, lines 142-148 for example. These details are not central to the understanding of ONI and could have been pushed to the appendix.
>
> The system design (which we will open source the code of) is a core contribution of the paper and we believe this is important to highlight. These engineering contributions significantly widen the range of settings where intrinsic reward methods can now be applied.
>
> > URL in the abstract does not work
>
> This is a placeholder. We will add a link to the code repo when the paper is published.
>
> > η is not rendered properly in Fig 5.2
>
> Thanks for pointing this out, we will fix this in the next version.
>
> We hope that we have addressed all your concerns through our answers and additional experiments, and that you will consider raising your score as a result. If there are any remaining concerns that stand between you and a recommendation of acceptance, please let us know as soon as possible and we will do our best to address them.

---

> > ### Comment · Reviewer_9m6i · 2024-11-27
> >
> > Thanks for your responses and clarifications.

---

> > > ### Author Response · Authors · 2024-11-28
> > >
> > > Thank you for acknowledging our response and raising your score. However, given our additional experiments, paper updates and responses to your questions above, we do not have clarity on what remaining concerns you have that would prevent you from recommending acceptance. Could you either elaborate on which concerns remain unaddressed, or raise your score to accept? We are happy to help address any remaining concerns you have.

---

> > > > ### Author Response · Authors · 2024-12-02
> > > >
> > > > Dear reviewer,
> > > >
> > > > Given that the discussion period ends tomorrow, could you kindly let us know if you have any remaining concerns that prevent you from recommending acceptance for our paper?
> > > >
> > > > Thank you again for your time and effort in reviewing our paper.

---

> > > > ### Comment · Reviewer_9m6i · 2024-12-03
> > > >
> > > > Thanks for your efforts. I believe that if the system design is a core contribution, these should be stated and presented distinctly from the conceptual contributions. What ONI stands for is yet to be mentioned in the paper.  As pointed out by my fellow reviewers, the general applicability of the work is still questionable. Overall, it would have been better to make the code available for the sake of transparency and reproducibility.

---

> ### Author Response · Authors · 2024-12-03
>
> Dear reviewer,
>
> Thank you for listing your remaining concerns. We would like to make sure that you have seen the relevant parts of the paper and our response above, in particular:
>
> - the system design and algorithmic design are currently in two separate sections (3.1 and 3.2)
> - we clarified above that ONI stands for **ON**line **I**ntrinsic Rewards, which we will specify in the updated paper
> - we clarified above that **we will open source the code** upon publication, ensuring full transparency and reproducibility.
>
> We think that these address 3 out of the 4 the concerns you have listed. If not, please let us know which parts are not satisfactory and we will be happy to elaborate.
>
> Concerning the general applicability: our use of LLMs currently limits us to settings with observations which are text-based or include some textual component. Out of the available such environments, we believe NetHack to be the most challenging - this is corroborated by the recent BALROG benchmark, which shows that out of 6 available text-based benchmarks, Nethack is the one where SOTA LLMs struggle the most (https://balrogai.com/).
>
> To run evaluations on non-textual visual benchmarks, one would need high-performing open VLMs. At the time of submission, these were not available. Since then, LLaMA3.2-V and Pixtral-12B have been released, and we are currently working on integrating them into our system. However, we do not believe we should be penalized for not having such results, since these open VLMs were not available at submission time.

---

### Official Review · Reviewer_4gbj · 2024-11-03

**Soundness:** 3
**Presentation:** 3
**Contribution:** 2
**Rating:** 6
**Confidence:** 3

**Summary:**

This work presents a distributed system and an online learning algorithm for developing a reinforcement learning policy that learns from an intrinsic reward function, which is trained/obtained using feedback from LLMs. The approach depends solely on online experience, removing the need for external datasets or source code, and is assessed in the NetHack environment, comparing its performance to that of the Motif algorithm.

**Strengths:**

1. The proposed research problem—scaling up LLMs to generate dense intrinsic rewards for online learning—is compelling and significant for advancing LLM agents' ability to tackle complex, long-horizon tasks in real-world applications without assumption of access to expert demonstration.

2. The paper demonstrates strong engineering effort, employing online learning with help of LLMs to train an agent with up to 2 billion environment steps, a notable advance compared to prior work that required either only a few iteration of online training or a large offline dataset.

3. The experimental design effectively supports the method’s claims. The chosen testbed is well-suited, as its long-horizon nature necessitates LLM-generated intrinsic rewards to achieve successful outcomes.

**Weaknesses:**

1. (main) The authors compare their work only with the Motif baseline algorithm. It would be beneficial to include comparisons with goal-conditioned reward design approaches as well. In the related work section, specifically in the goal-conditioned reward design subsection, the authors list relevant studies but do not critique their limitations or highlight the advantages of their approach. Adding these comment, as done in other subsections, would strengthen the discussion.

2. In Figure 5.4, it is surprising that a higher number of annotations does not improve performance. The author might conduct an ablation study to assess the relationship between annotation volume and performance—reducing annotated data until a performance drop is observed. This may reveal that fewer labeled data points are sufficient for training the reward function, potentially reducing computational resources.

3. Using HTTP for communication between the LLM server and the annotation process may incur high communication costs. Although the authors claim minimal overhead in line 189, an experiment to demonstrate this would provide clearer evidence.

**Questions:**

I have outlined all my concerns in the weaknesses section. I would be happy to discuss further if the authors can address my questions.

---

> ### Author Response · Authors · 2024-11-26
> **Thank you for the review**
>
> We thank the reviewer for the constructive feedback and respond to your questions below.
>
> **Comparison with goal-conditioned baseline:**   This is a good suggestion, we have added such a baseline based on [1]. Please see our common response where we discuss this in detail, and the experiments we have added to the paper in Section 5.1 and 5.2 (these are marked in red).
>
>
> **Reducing the Amount of Annotated Data:**  We have added this ablation and included the results in Figure 5.5. We also discuss this point in the common response.
>
> **Use of HTTP for Communication:**  It is pretty standard that API-based LLMs use HTTP, including OpenAI, Claude, LLaMA, Mistral.  Using HTTP communication allows us to host the LLM on a separate server node, which is most flexible and allows scaling to very large size LLMs and many nodes. That being said, our architecture also allows us to host LLM on the same node if the computational resources allow, which further reduces communication costs. We have added a sentence clarifying this in the updated paper.
>
> We hope we have addressed your concerns, and that you will consider raising your score. If not, please let us know what remaining concerns you have and we will do our best to address them.
>
> [1] Guiding Pretraining in Reinforcement Learning with Large Language Models, ICML 2023

---

> > ### Comment · Reviewer_4gbj · 2024-11-26
> >
> > Thank you for the detailed response to my earlier feedback. I have two remaining questions that I'd like addressed before providing my final review:
> >
> > 1. Could you revise the Goal-conditioned Reward Design paragraph in the related work section to better emphasize the limitations of prior work and clarify the advantages of your proposed approach?
> >
> > 2. Why did you choose HTTP over MPI or NCCL for communication?

---

> > > ### Author Response · Authors · 2024-11-26
> > >
> > > Thank you for the quick response. To answer your questions:
> > >
> > > 1. We have revised the paragraph in the Related Work section to make the limitations of prior work which requires LLM calls at each observation (such as ELLM) more explicit (this is marked in red). Please let us know if this is satisfactory, we are happy to update further if needed.
> > >
> > > 2. If the LLM server uses multiple GPUs for inference (for example, by using pipeline parallelism or tensor parallelism), the communication between these GPUs is typically implemented using NCCL. Many libraries such as VLLM use this under the hood. HTTP is instead used for communication between the LLM process as a whole and external processes. In our case, this is in Sample Factory, but it could be any user-controlled process more generally. So in short, we do use both HTTP and NCCL, but the NCCL is handled internally by the LLM serving library (fastchat/VLLM in our case). To illustrate schematically, if we have 2 GPUs serving the LLM, the data flow would be:
> > >
> > >
> > > ```
> > > 	 (APPO process)     ← HTTP →    (GPU_1 ← NCCL → GPU_2)
> > > ```
> > >
> > >
> > >
> > > We hope this clarifies your remaining questions. We have updated the paper to reflect 1). Please let us know if there are any remaining questions and we will do our best to answer them.

---

> > > > ### Author Response · Authors · 2024-12-02
> > > >
> > > > Thank you again for your time reviewing our paper and your earlier response. Given that the discussion period ends tomorrow, could you kindly let us know if our response above adequately answers your two remaining questions, and if so, update your review? If not, please also let us know and we will do our best to address any remaining concerns.

---

> > > > > ### Comment · Reviewer_4gbj · 2024-12-02
> > > > >
> > > > > Thank you for your clarification; it addressed most of my concerns. Overall, I appreciate the problem tackled in this paper—using large-scale online RL learning with intrinsic rewards generated by LLMs. The paper presents a simple yet effective step toward solving this task, opening up opportunities for future work. I will vote for acceptance while encouraging the authors to test on a broader range of benchmarks.

---

### Author Response · Authors · 2024-11-26
**Common response**

We are grateful to all reviewers for their detailed and constructive feedback on our paper, and we are encouraged to see that reviewers find:


- Our proposed research problem — utilizing LLMs to generate online dense intrinsic rewards — is compelling and significant for advancing LLM agents' ability to tackle complex, long-horizon tasks in real-world applications (Reviewer 4gbj, 9m6i, Cv7N, C2KC)
- ONI eliminates the need for external datasets and instead building rewards from the agent’s own experience, ONI offers a scalable approach that could adapt to a wide range of tasks (Reviewer 4gbj, C2KC)
- Our paper demonstrates strong engineering effort, the asynchronous design keeps the policy training process uninterrupted, maintaining high throughput and efficiency throughout (Reviewer 4gbj, C2KC)



We have added a number of experiments in response to reviewer comments. We summarize these here, and provide responses to individual reviewers below.

* **Additional Baselines:** Reviewers **4gbj**, **9m6i** and **C2KC** all asked to see additional baseline comparisons, specifically to a goal-conditioned baseline. First, we would like to note that the specific LLM-based baselines suggested (such as the ELLM method of [Du et al, 2023]) are not computationally feasible in our setting, since they require an LLM call for each observation —and observations are in the billions here. Therefore, we implemented a modified version which does scale to our setting, where we replace the LLM embedding with a bag-of-words embedding using FastText, and also add an episodic term to the intrinsic reward. This reveals several interesting findings:
    * Directly using ELLM-BoW as in Du et al. 2023, without the episodic term, completely fails, highlighting that this episodic term is crucial. We present the results in Appendix C.6.
    * With our modification, ELLM-BoW provides strong performance on the Oracle and Staircase tasks, almost matching Motif and ONI without using an LLM. This shows that an LLM may in fact not be needed for these particular tasks, and simpler text processing can be sufficient, see Figure 5.1.
    * Although ELLM-BoW matches ONI on Oracle and Staircase, it does worse than ONI on tasks where more sophisticated goal understanding is required. For example, we tested ELLM-BoW and ONI with two opposite goals: “collect gold, but do not kill monsters” vs “kill monsters, but do not collect gold”. ELLM-BoW fails to understand the semantic differences and results in agents with similar behavior, while ONI is able to capture the semantic differences and the obtained agents have different focuses. This stems from the nature of the bag-of-word embedding, which unfortunately is the only scalable variant, as we mentioned before. This shows that the LLM is indeed needed as the goal complexity increases. We also tried more sophisticated sentence transformers instead of bag-of-words embeddings, but these hurt the throughput too much (a 20-100x slowdown).
    * The takeaway from these experiments is that ONI is the only method which can i) scale to high-throughput settings, ii) operate purely online, without offline data, and iii) handle complex goal specifications which require an LLM.

* **Annotation Ablation:** Reviewers **4gbj**, **9m6i**, **Cv7N** all asked for more details about how performance is affected as we decrease the annotation budget. We have added an additional set of ablations where we significantly reduce the number of annotations (by a factor of 0.1 and 0.01). We find that ONI-retrieval significantly drops in performance, whereas ONI-classification remains relatively robust. This provides more evidence for the utility of the parametric reward model, since it can generalize from a few annotations and reduce the need for annotation.

We have also made some changes to the ranking algorithm: specifically, we found that with additional hyperparameter tuning, the differences between the offset and threshold variants become small. Therefore, we decided to stick with the threshold variant in order to introduce fewer changes compared to offline Motif.



We believe have addressed all the questions raised by reviewers with additional experiments or thorough clarifications via separate responses to each reviewer. Please let us know if you have other questions, and we are happy to discuss them.

---

### Author Response · Authors · 2024-11-26
**Common Response II -- System**

Dear Reviewers,

Reviewer Cv7N brought up one concern that our system contribution is trivial, as it seems a simple add-on to sample factory. After some thoughts, we decided to write a common response for clarification, as we think the system contribution is important and need to be discussed.

To start, even though we are not implementing Sample Factory from scratch, the engineering effort for adding this async LLM sever and async reward model training is nontrivial. There are multiple difficulties that need careful design and treatment, otherwise the training would not scale or crash. Here are a few example questions we need to answer through the development of our codebase:

- **How to setup the LLM server? Should we use a child process or setup a remote server?** Spawning a child process and invoke a VLLM instance is the easiest solution, however, this means the APPO code and LLM code need to run on a same node, which prevents the use of large-size LLM e.g. 70B LLaMa that requires a single node's compute itself. To make the framework more general and enable future extensions to large VLMs, we chose the latter solution, which is way more difficult.

- **In which thread should we update the reward model? How to keep the LLM's annotation throughput as high as possible?** If we use the feedback processing thread that talks with LLM to update the reward model, it stalls the communication with LLM server and hurts the annotation throughput; but updating the reward model in the training thread will quickly overfit the small set of annotated data in the beginning. Therefore, we choose to tolerate the paused communication with LLM server for a short period of time -- updating the reward model in the feedback processing thread until we collect 25k annotations.

- **How to enable thread-safe annotation data reading & writing, thread-safe reward model training and thread-safe checkpointing?**
Due to the nature of async training, there are 2+ threads reading & writing the annotation hash table, 2+ threads updating the reward model & evaluating it to obtain intrinsic rewards for policy training. We need to make these operations thread-safe. Also, as the training takes up to 25+ hours, we need to save our models periodically outside the training/feedback processing thread. Unfortunately, neither torch.save() nor python pickle is thread-safe. Our system needs multiple thread locks that cover these 3 aspects, we need to carefully design & code them, also preventing deadlock.

- **How to enable experiments on a Slurm managed cluster using such remote framework? How to enable debugging in an asynchronous environment?** This might sound easy, but in fact, it is more challenging than we initially thought, especially if you want to manage so many experiments where training an agent for a single NetHack task takes 25+ hours, with 2B env steps. We have 5 tasks (1 dense reward, 3 sparse reward, 1 reward free), 3 ONI methods, and 2 baselines (Motif, ELLM-BoW). We ended up with using heterogeneous jobs (https://slurm.schedmd.com/heterogeneous_jobs.html) that launch LLM server first, then launch the main training code with the URL address of the LLM server. As mentioned before, managing such large scale experiments is non-trivial, the volume of the experiments presented in our paper is large. We need to carefully code the logging of the LLM server and APPO server for debugging.

Finally, for debugging in an asynchronous environment, the first set of questions are **How to attach a debugger to a python subprocess? Will it stall the computation for other subprocesses?** -- more questions are coming. The answers to those questions, at least, are not obvious to us in the beginning. We are using ForkedPdb suggested here: https://stackoverflow.com/questions/4716533/how-to-attach-debugger-to-a-python-subproccess, which sometimes works but is also hard to use as other subprocesses will flush the IO. Even now, we don't have a perfect solution for python asynchronous training. (If reviewers have good suggestions, please do let us know.)

To summarize, we want to let reviewers know the engineering difficulty & effort we put are both nontrival. We have spent a lot of time to make sure the code quality is high. We did not discuss this too much in the paper, in order to put the paper's focus on the intrinsic reward design. We are working on refining the code and open source it later.  We strongly believe the system work is an important and valuable step, as it paves the road to later research in this direction. Therefore, we also (really!) hope and appreciate if you can see the long-term value of our work.

Best,
Paper 5187 authors

---

### Meta-Review · Area_Chair_Hkg6 · 2024-12-24

**Metareview:**

This submission presents a framework for incorporating LLMs into an online RL setup by learning an intrinsic reward function concurrently with the policy. The idea is to asynchronously gather LLM annotations and distill them into a learned reward model, enabling sparse-reward problems to be solved more effectively. The authors compare three approaches (retrieval, classification, ranking) in the NetHack environment and claim these methods remove the need for pre-collected datasets while achieving strong performance.

The general idea of using LLM-based annotations for dense intrinsic reward design is interesting. However, there were concerns about lack of novelty and incremental contributions. The underlying innovations in system design are minimal compared to prior distributed RL architectures, and the method’s reliance on standard RL building blocks leaves little conceptual novelty. This is fine if empirical support outweighs the limitations. This work is narrowly evaluated on the NetHack environment. While challenging, this environment has a specific observation space.  It is not clear whether the proposed approach generalizes to more widely used RL benchmarks. This is its biggest limitations and why it didn't end up getting better scores.

**Additional Comments On Reviewer Discussion:**

Given the concerns raised during the review process—particularly the lack of broader applicability, depth of innovation, and a convincing demonstration of benefits on varied tasks—I recommend rejecting this submission at this point.

The authors should strengthen their core claim by, 1) showcasing broader empirical validation (non-text or more complex domains), 2) refining how LLM bias and annotation overhead are handled, and 3) better clarifying the unique algorithmic contributions beyond standard asynchronous frameworks. I would encourage the authors to expand upon the work and improve it based on the feedback received during this rebuttal process.

---

### Decision · Program_Chairs · 2025-01-22

Reject